# Occurrence of Aflatoxins in Edible Vegetable Seeds and Oil Samples Available in Pakistani Retail Markets and Estimation of Dietary Intake in Consumers

**DOI:** 10.3390/ijerph18158015

**Published:** 2021-07-29

**Authors:** Muhammad Waqas, Shahzad Zafar Iqbal, Ahmad Faizal Abdull Razis, Wajeeha Pervaiz, Touheed Ahmad, Sunusi Usman, Nada Basheir Ali, Muhammad Rafique Asi

**Affiliations:** 1Department of Applied Chemistry, Government College University Faisalabad, Faisalabad 38000, Pakistan; mwaqasgujjar028@gmail.com (M.W.); wajeeha.309@gmail.com (W.P.); toheedipe@gmail.com (T.A.); 2Natural Medicines and Products Research Laboratory, Institute of Bioscience, Universiti Putra Malaysia, Serdang 43400, Selangor, Malaysia; usunusi.bch@buk.edu.ng; 3Department of Food Science, Faculty of Food Science and Technology, Universiti Putra Malaysia, Serdang 43400, Selangor, Malaysia; nada44basher@gmail.com; 4Food Toxicology Lab, NIAB, Faisalabad 38000, Pakistan; asimuhammad@yahoo.co.uk

**Keywords:** AFB_1_, AFs, vegetable seeds, vegetable oils, dietary intake

## Abstract

Aflatoxins (AFs) are secondary metabolites toxic to humans as well as animals. The environmental conditions, conventional agricultural practices, and illiteracy are the main factors which favor the production of AFs in food and feed. In the current study 744 samples of vegetable seeds and oils (soybean, sunflower, canola, olive, corn, and mustard) were collected and tested for the presence of aflatoxin B_1_ (AFB_1_) and total AFs. Liquid-liquid extraction was employed for the extraction of AFs from seeds and oil samples. Reverse phase high performance liquid chromatography equipped with fluorescence detection was used for the analysis. The results have shown that 92 (56.7%) samples of imported and 108 (57.0%) samples of local edible seeds were observed to be contaminated with AFs. All samples of edible seeds have AFB_1_ levels greater than the proposed limit set by the European Union (EU, 2 µg/kg) and 12 (7.40%) samples of imported seeds and 14 (7.40%) samples of local seeds were found in the range ≥ 50 µg/kg. About 78 (43.3%) samples of imported edible oil and 103 (48.3%) sample of local edible oil were observed to be positive for AFs. Furthermore, 16 (8.88%) and six (3.33%) samples of imported vegetable oil have levels of total AFs in a range (21–50 µg/kg) and greater than 50 µg/kg, respectively. The findings indicate significant differences in AFs levels between imported and local vegetable oil samples (t = 22.27 and *p* = 0.009) at α = 0.05 and a significant difference in AFs levels were found between vegetable seeds and oil samples (t = −17.75, *p* = 0.009) at α = 0.05. The highest dietary intake was found for a local sunflower oil sample (0.90 µg/kg/day) in female individuals (16–22 age group). The results have shown considerably high levels of AFB1 and total AFs in seeds and oil samples and emphasise the need to monitor carefully the levels of these toxic substances in food and feed on regular basis.

## 1. Introduction 

In the current century, edible vegetable oils are preferred over animal oils during frying of food or for use in food handling industries due to health issues. Important human nutrients such as energy, vitamins, fat soluble and essential fatty acids are supplied by oils [1]. Consumers use edible oils in daily life because they have functions in preventing arteriosclerosis and reducing blood lipids [2,3]. Increasing awareness and the health benefits of using vegetable oils have boosted their demand and consumption worldwide, especially in developed countries [4]. Worldwide, the most highly popular oils are maize oil, olive oil, peanut oil and sunflower oil, and the consumption rates of these oils is rising. The U.N. Food and Agriculture Organization (FAO) [5] has estimated the total utilization of oil and fats worldwide in 2020/21 was 244.8 million tons. China is the leading producer of oilseeds, with some 2.63 million tons, followed by Pakistan and Malaysia, with 55.8 thousand tons and 20.2 thousand tons, respectively [6]. According to a report, from 2001 through 2011, the projected mean global per capita consumption of vegetable oils was 10.71 kg, which is 1.24 kg higher than vegetable oil consumption from the previous decade [7]. The per capita edible oil consumption of Pakistan is about 17 kg, and in 2017 the total consumption of oil and fats was about 4.41 million tons. Pakistan imported about 2.7 million tons of edible oil in 2015, which rose to 3.05 million tons in 2017 [8]. The increasing trend in the utilization of vegetable oil has attracted the attention of regulatory authorities and agencies to check on its safety and quality [9]. The processing and handling of oil during its production, packaging, transportation and storage could cause contamination, furthermore with innovation in industrial processing, traditional agricultural practices, environmental pollution and climatic conditions may cause the presence of new toxic residues in edible oils [10].

The contamination of food and food products with aflatoxins (AFs) is a global food safety concern [4]. AFs are recognized as dangerous and toxic natural compounds. The AFB_1_ subclass of AFs is recognized as the most toxic and carcinogenic [11,12,13]. Fungi like *Aspergillus flavus*, *Aspergillus nomius*, and *Aspergillus parasiticus* are the primary producers of aflatoxins [14]. These fungi can contaminate various food and food products, e.g., vegetables, fruits, cereals, spices, and cattle feed [15,16]. Considering its toxicity, AFB_1_ has been categorized as a group 1 carcinogen by the International Agency for Research on Cancer [17], and it mainly affects the liver [18]. Geographically, Pakistan is placed in the list of tropical countries of the world, and therefore the climatic conditions may be favorable for fungal metabolite production [19]. High temperature and humidity levels are conducive for the growth of aflatoxigenic fungi. High levels of toxic compounds like AFs in foods or vegetables could be easily transferred to final edible oil foodstuffs from the seeds [20]. Furthermore, the pre-harvest and post-harvest stages, primarily due to inadequate storage conditions, are mainly responsible for contaminating food products due to fungi [21,22]. The climatic conditions are considered significant too for the production of fungi in food products [23,24]. There were numerous surveys worldwide that account for the contamination of AFs in vegetable oils [4,20,25,26,27,28]. In Pakistan, no previous studies have conducted on the occurrence of AFs in edible oils. However, a high incidence of AFs was reportedly present in feed samples (cereal products) [29] and more recently in animal feed [30]. In a previous study, a high incidence of AFs, i.e., in 180 (43.4%) samples of edible seeds from the winter season and 122 (33.4%) samples from the summer season was found [31].

Therefore, this study was designed to investigate the levels of AFB_1_ and total AFs in edible vegetable oils, to compare the levels with EU recommended limits, and to estimate the possible dietary intake in the local population. The findings of the current research will help to understand the toxicity of AFs in vegetable oils, to generate data about their incidence, and help to implement strict regulations in Pakistan. 

## 2. Materials and Methods

### 2.1. Sample Collection

Three fifty-one (351) samples of edible seeds and 393 samples of edible oilseeds (sunflower, soybean, canola, olive, corn and mustard) were gathered from markets, superstores, and farmers from the central cities of Punjab, Pakistan during May 2019 to August 2019. The imported samples were named those imported from other countries, and local ones were produced locally. These imported sunflower seeds samples were collected from five different brands, seven different brands for soybean, eight different brands of canola, olive and corn samples and four brands for mustard in the main cities of Punjab in Pakistan (Lahore, Faisalabad, Gojira and Islamabad) and subsequently the oil was extracted (Soxhlet apparatus) from these samples and labeled accordingly as shown in Figure 1. The sample size was maintained from 1 kg to 5 kg, each. A simple random methodology (each portion or lot has equal chance to be included) was used for collecting edible seed samples from farmers, markets, and superstores. The gross samples were collected by hand and then homogenized and properly labelled was done. The size of each brand must ensure to be greater than *n* = 20, to be assumed to represent normal distribution. All samples were stored in polythene zip bags or airtight plastic bottles. All the samples of seeds and oil were stored at room temperature (25–30 °C) in the dark.

### 2.2. Chemicals and Reagents 

The standards of AFB_1_, AFG_1_ (2 μg/mL in acetonitrile) and AFB_2_ and AFG_2_ (0.5 µg/mL in acetonitrile), methanol, acetonitrile (HPLC grade), hexane, sodium chloride, chloroform, anhydrous sodium sulfate, dichloromethane, HCl and trifluoroacetic acid (TFA) were obtained from Sigma-Aldrich (Steinheim, Germany). Furthermore, double-distilled water (Millipore, Bedford, MA, USA) was used throughout the analysis.

### 2.3. Extraction of Aflatoxins from Seed Samples

The extraction of AFs from edible seed samples was performed following the literature methods [32,33]. The sample (25 g) was mixed in methanol:water (55:45 *v*/*v*, 125 mL), hexane (100 mL) and NaCl (2 g) and homogenized using an orbital shaker for 15 min. Then Whatman No.1 filter paper was used to filter the solution. After filtration, the filtrate was left for 30 min to form two phases. The lower 25 mL of the aqueous methanol phase was transferred into a separatory funnel, and 10 mL of chloroform was added. This process was repeated three times in the separatory funnel. Two layers formed, and the chloroform layer was drained into a 250 mL beaker and dried using anhydrous sodium sulfate. Then, the final solution was evaporated over a water bath to near dryness.

### 2.4. Extraction of Aflatoxins from Oil Samples

The method for AFs extraction in edible oil samples was carried out as explained by the AOAC Official Method 2013.05 [34]. With some modifications, 50 mL of oil sample was mixed in 250 mL of methanol-water (55–45 *v*/*v*) with the addition of 50 mL of 0.1 N HCl and centrifuged at a speed of 4500 rpm. The mixture was filtered with Whitman filter paper, and 50 mL of the filtrate was placed in the separatory funnel and mixed with 50 mL of 10% NaCl and 50 mL of hexane and shaken vigorously for 30 s. The aqueous lower layer was drained into another separatory funnel, and 3–25 mL of dichloromethane was added and the mixture shaken vigorously and allow it to stand for 5 min. Then the dichloromethane layer was collected and evaporated on a water bath to dryness. 

The derivatization of both seed and oil samples were carried out using 100 µL of TFA in dried oil or seed samples and vortex for 30 s. Then the sample was left for 5 min in a dark place. Finally, 400 µL mixture of acetonitrile-water (1:9 *v*/*v*) was added to the vials, and 20 µL of solution was subjected for HPLC analysis.

### 2.5. HPLC Conditions

The research was conducted to investigate the incidence of AFs on a Model-LC-10A HPLC instrument (Shimadzu, Kyoto, Japan) equipped with a C_18_ column (250 mm × 4.6 mm, 5 µm) (Discovery, HS, Bellefonte, PA, USA) equipped with a fluorescence detector (Model RF-530). The polar isocratic reverse mobile phase consisted of acetonitrile, water and acetic acid (50:40:10 *v*/*v*/*v*) pumped at a flow rate of 1 mL/min. The emission wavelength (440 nm) and excitation wavelengths (365 nm) of the fluorescence detector were set before the analysis. 

### 2.6. Dietary Intake Estimations

The dietary intake analysis was performed following the method depicted by Iqbal et al. [35], the estimated daily intake (EDI) is calculated as:$$\mathrm{Dietary}\;\mathrm{intake}\;\left(µg/\mathrm{kg}/\mathrm{day}\right)\;=\frac{\mathrm{Consumption}\;\mathrm{of}\;\mathrm{oil}\;\left(\mathrm{mL}\right)\;\times \;\mathrm{Mean}\;\mathrm{levels}\;\mathrm{of}\;\mathrm{total}\;\mathrm{AFs}\;\left(µg/\mathrm{kg}\right)}{\mathrm{Average}\;\mathrm{weight}\;\left(\mathrm{kg}\right)\;\mathrm{of}\;\mathrm{individuals}}$$

The intake data was obtained by administering a food frequency questionnaire (available in the Appendix A) to 645 participants of which 509 responded and returned the information about their oil use for different cooking food items. The questionnaire examined every aspect of oil consumption, dietary supplements, and utilization. The bodyweight of the participants was 61 ± 11 kg. Written consent was obtained from each participant who were assured their information would not be made public. All ethical guidelines have been adopted during questionnaire completion. The participants were consumers and their dietary habits and seasoning effects was not considered, which might affect the results. 

### 2.7. Statistical Analysis

The findings of the current research were analyzed statistically and the results presented as mean ± standard deviations. The calibration curves of seven-points were constructed for each AFs using simple linear regression/correlation analysis, and coefficient of determination and straight-line equations were calculated. The significant difference among the levels of AFs in imported and local samples and edible seeds and oil samples were verified applying paired *t*-test (α = 0.05) SPSS (IBM, Chicago, IL, USA). The method was evaluated in terms of linearity, reproducibility, repeatability, recovery analysis, detection limits (LOD), and the limit of quantification (LOQ). Three known concentrations of AFB1 and aflatoxin G1 (AFG1) (1, 6, and 10 µg/kg) and aflatoxin B2 (AFB2), and AFG2 (1, 4, and 8 µg/kg) were added in a mixture of samples of edible oils (all five oils samples with equal volume), which have detection levels of all four AFs < LOD.

## 3. Results and Discussion

### 3.1. Quality Control Parameters

The mean recovery values (each in triplicate) ranged from 74.5% to 96.5%, with a relative standard deviation (RSD) that varied from 9% to 21.5%, as shown in Table 1. A calibration curve (seven points) was constructed to confirm the linearity of each AFs’ response, i.e., for AFB_1_ and AFG_1_ (1, 10, 20, 60, 80, 100, 140 µg/kg) and AFB_2_ and AFG_2_ (1, 4, 8, 16, 20, 25 and 30 µg/kg). The curves were linear, with a coefficient of determination (R^2^) ≥ 0.99. The detection limits (LOD) of AFB_1_ and AFG_1_ were 0.08 µg/kg, and LOQ was 0.24 µg/kg. However, the LOD and LOQ for AFG_2_ and AFB_2_ were 0.07 and 0.21 µg/kg, respectively. The repeatability and reproducibility have also been calculated. In the previous study, the linearity range for AFB_1_ and AFG_1_ was 1 to 80 µg/kg and 0.5 to 12 µg/L for AFB_2_ and AFG_2_, in agreement to current study. The LOD and LOQ were 0.04 and 0.12 µg/kg for AFB_1_ and AFG_1_ and 0.6 and 0.18 µg/kg for AFG_2_ and AFB_2_, respectively [36]. The peaks of individual AFs are separated quite efficiently. The standard chromatogram showing the retention time of four individual AFs (Figure 2A); The natural incidence of four AFs in sunflower sample (Figure 2B); and the presence of individual AFs in mustard seed sample (Figure 2C) are represented. 

### 3.2. Occurrence of AFs in Edible Seeds and Oil Samples

The results have revealed that 92 (56.7%) samples of imported and 108 (57.0%) samples of local edible seeds were observed to be contaminated with AFs. The maximum average amount of AFB_1_ and total AFs in local soybean seeds were 21.01 ± 4.70 and 36.37 ± 6.10 µg/kg, respectively. The amount of 13.29 ± 3.50 and 20.42 ± 5.20 µg/kg was found for AFB_1_ and total AFs in imported sunflower seed samples, respectively, as shown in Table 2. Furthermore, all edible seeds samples have levels of AFB_1_ greater than the proposed EU limit (i.e., 2 µg/kg). On the other hand, 12 (7.40%) samples of imported and 14 (7.40%) samples of local seeds were found in the range ≥ 50 µg/kg, as shown in Figure 2A. About 393 samples of edible oil, of which 180 samples were imported, and 213 samples were local, were investigated for the incidence of AFB_1_ and total AFs, as shown in Table 3. The 78 (43.3%) samples of imported and 103 (48.3%) samples of local edible oils were positive with AFs. The maximum average amount of AFB_1_ (14.29 ± 2.51 µg/kg) and total AFs (25.61 ± 7.50 µg/kg were documented in local samples, and the amount of 6.94 ± 1.90 and 12.71 ± 4.30 µg/kg were documented in imported soybean oil samples, respectively. About 46.6% samples of sunflower oil have levels of total AFs in a range of (21–50 µg/kg) as shown in Figure 3A. Furthermore, 66.6% samples of soybean oil produced locally have concentrations of total AFs in a range of ≥50 µg/kg, as shown in Figure 3B. There existed a higher level of AFs in locally produced vegetable oil samples compared to imported samples (t = 22.274 and *p* = 0.009) at α = 0.05. A significant difference in levels of AFs in seeds compared to oil samples were found (t = −17.75, *p* = 0.0009) at α = 0.05.

High incidence and higher levels of AFs than the current study’s findings were reported from Iran by Beheshti and Asadi [37]. They observed 111 (64%) out 173 samples of sunflower and safflower were found to be contaminated with AFB_1_ and 103 (83.7%) samples of safflower seeds with a mean of 2.81 to 0.44 ng/g and eight (16%) samples of sunflowers with a mean of 40.68 ng/g were contaminated with AFB_1_. However, only five and two samples of sunflower and safflower seed were levels higher than the EU limit (2 µg/kg), respectively. Similarly, from the same geographical region, i.e., from Sri Lanka, Karunarathna et al. [4] have analyzed 59 vegetable oil samples (43 imported and 16 local) from seven different categories (coconut, palm, olein, sunflower, olive, sesame, soybean and corn soil). They documented that 12 (37.5%) out of 32 coconut samples were observed to be contaminated with AFs. They documented that AFB_1_ and total AFs ranged from 2.25 to 72.70 µg/kg and 1.76 to 60.92 μg/kg, respectively, and two out of 12 oil samples with levels that exceeded the EU’s high permitted limit of 2 µg/kg, for AFB_1_. Mohammed et al. [38] have studied 40 samples of sunflowers seeds and 21 samples of unrefined oils of sunflowers from Tanzania, and found that six (15%) samples were discovered to be infected with AFB_1_, ranging from LOD to 218 ng/g, comparatively higher than the findings of the present study. Only three samples have levels greater than the Tanzanian Bureau of Standards (TBS) and EC/EU permissible limit.

However, studies documented a high occurrence of positive samples with AFs contamination but with low AFs concentrations. Banu and Muthumary [25] have shown that 10 (43.4%) out of 23 samples of sunflower oil from India, were observed to be contaminated with AFB_1_, and all refined oil samples have levels lower than LOD. Ferrracane et al. [26] have studied virgin olive oil samples from Europe, i.e., from Italy and Morocco, and found that only three (10%) out of 30 samples of olive oil was found to be contaminated with AFB_1_ along with ochratoxin A; ranging between 0.54 to 2.50 ng/g. Mariod and Idris [27] documented that 54.8% of samples of groundnuts and 14.5% (out of eight) sunflower oil samples from Sudan, were found to be contaminated with AFB_1_. Nabizadeh et al. [39] have examined 97 edible oil samples of six categories (olive, sunflower, canola. blends, frying, unrefined olive oil). They observed that 98% of samples showed levels of AFB_1_ lower than LOD and all the positive samples with AFs were within the EU regulations 20 µg/kg. In Pakistan, Shar et al. [40] have observed the natural occurrence of AFB_1_ in 110 samples of cotton seeds and fount the maximum level of AFB_1_ in cottonseed cakes, i.e., 89 µg/kg. 

Edible oil and groundnuts are considered important cash crops in many parts of the world. However, due to the adoption of old conventional agricultural practices and illiteracy among farmers and traders regarding toxigenic fungi, makes the safety and quality of crops questionable. Edible oils (olives, sunflower, coconut) are often stored for long periods of time under non-ideal conditions e.g., in contact with the ground, moisture, in jute bags, etc. The extended storage period promotes the growth of molds, favoring toxicogenic mold colonization according to Yassa et al. [41]. Environmental conditions like drought might affect the crops during the preharvest phase and could produce fungi like *Aspergillus*. That is why it becomes unfit for human or animal consumption [42]. Under favorable conditions of temperature and humidity, these fungi grow on certain foods and feed products and thus produce secondary metabolites like mycotoxins. High humidity and temperature should be controlled, which is a severe problem in tropical countries, like Pakistan, to avoid aflatoxigenic fungi growth [20]. Furthermore, the variation in results of AFs in food depends on various factors, like the analytical methods used, environmental conditions, crops and harvesting practices. Adopting good harvesting practices, and good storage practices might minimize the presence of aflatoxigenic fungi in food and products. 

### 3.3. Estimation of Dietary Intake in Sunflower Oil Samples

The estimation of dietary intake of total AFs in sunflower oil samples in different age groups of male and female individuals is presented in Table 4. Sunflower oil is the most consumed vegetable oil in Pakistan, and therefore a more realistic approach was to estimate dietary intake from this oil. The highest dietary intake was found in a local sunflower oil sample (0.90 µg/kg/day) in female individuals (16–22 age group), followed by the dietary intake of 0.69 µg/kg/day body weight in the male group (16–22 age group). The results have shown that high dietary intake values were found in both male and female individuals of the 16–22 years of age group. The local sunflower oil samples have shown the highest dietary intake levels in both male and female individuals. No comparative data to correlate the results of the present study was found. The high dietary intake levels of total AFs consuming sunflower could cause severe effects on the health of consumers, because the country already has insufficient health facilities. However, neglecting the dietary patterns, seasons and traditional eating habits of participants might alter the results of dietary intake assessments.

## 4. Conclusions

The study documented considerably high AFB_1_, and total AFs in vegetable seeds and oil samples from imported and locally produced samples. The results might drastically affect the consumption of locally grown vegetable oils. Regular monitoring of food and feed samples and organizing workshops for farmers, traders, and exporters might help create awareness about the toxic nature of AFs. Currently, the Punjab Food Authority has started to monitor food safety and quality, which is a good initiative. 

In the end the following recommendations could be helpful for farmers, traders, exporters, and consumers. It is mostly observed that during transportation or storage, the seeds were stored in jute bags, which may absorb moisture from the environment and provide favorable conditions for fungus formation in food. Monitoring the moisture and humidity levels are essential during transportation and storage. Furthermore, the seeds should be stored in polyethylene bags. More fungal-resistant crop cultivars should be preferred. In the future, more comprehensive research as a function of environmental conditions (moisture and humidity) is adequate to understand the fundamental questions of fungal growth and mycotoxin production.

## Figures and Tables

**Figure 1 ijerph-18-08015-f001:**
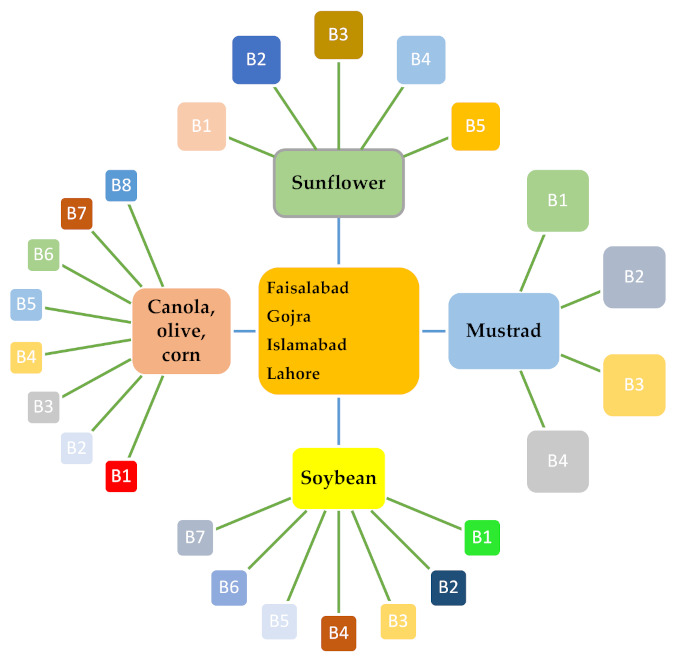
Edible seeds and oils samples distribution with respect to different brands.

**Figure 2 ijerph-18-08015-f002:**
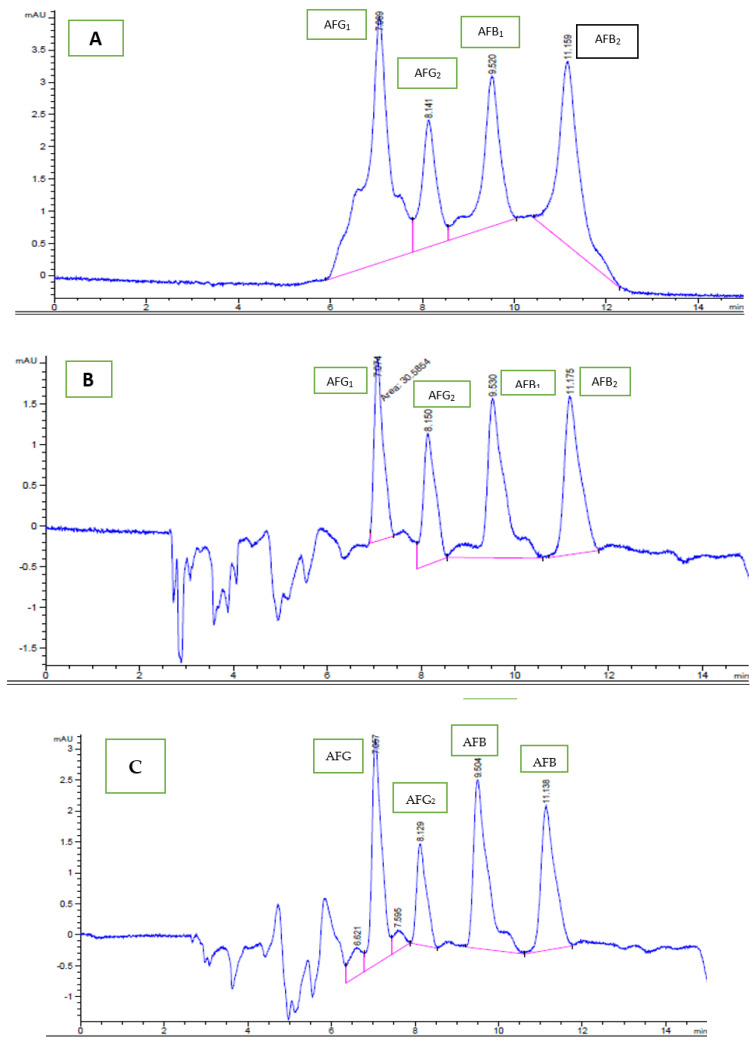
The presence of standard of all four AFs (**A**); the natural occurrence of AFs in sunflower sample (**B**); The presence of AFs in mustard seed sample (**C**). Pink and blue lines are chromatogram, pink lines shows baseline.

**Figure 3 ijerph-18-08015-f003:**
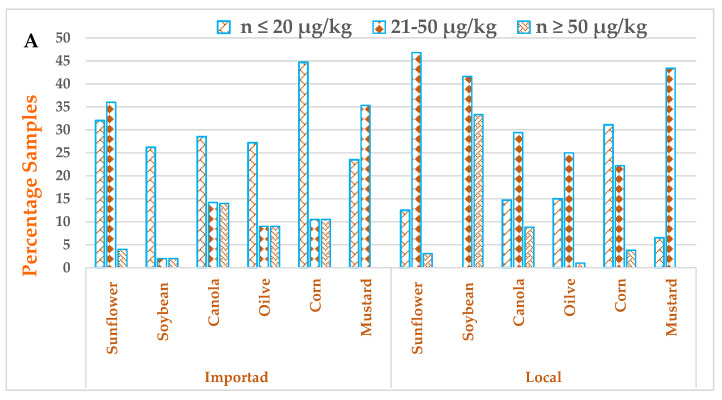
(**A**): The percentage of samples having total levels of AFs in (*n* ≤ 20 µg/kg), (21–50 µg/kg) and (*n* ≥ 50 µg/kg) in imported and local vegetable seed samples. (**B**): The percentage of samples having total levels of AFs in (*n* ≤ 20 µg/kg), (21–50 µg/kg) and (*n* ≥ 50 µg/kg) in imported and local vegetable oil samples.

**Table 1 ijerph-18-08015-t001:** Validation parameters of aflatoxins in edible seeds and oil samples.

Mycotoxins	Fortified Level (µg/kg)	Recovery %	RSD %	Retention Time (min)	R^2^	LOD (µg/kg)	LOQ (µg/kg)	Precision	
								Repeatability (*n* = 5) RSD%	Reproducibility (*n* = 5) RSD%
AFG_1_	1	84.5	9.0	7.074 ± 0.017	0.9910	0.08	0.24	14	11
	6	91.4	13.5						
	10	90.5	14.0						
AFB_1_	1	82.5	12.0	9.530 ± 0.026	0.9920	0.08	0.24	18	12
	6	94.5	10.0						
	10	96.5	21.5						
AFG_2_	1	74.5	18.5	8.150 ± 0.021	0.9980	0.07	0.21	19	10
	4	89.4	11.5						
	8	91.5	16.5						
AFB_2_	1	78.5	12.5	11.175 ± 0.037	0.9985	0.07	0.21	13	14
	4	94.6	11.1						
	8	90.5	14.0						

RSD = relative standard deviation, LOD = limit of detection, LOQ = limit of quantification; R^2^ = coefficient of determination; Repeatability and reproducibility are given as mean percent RSD (%).

**Table 2 ijerph-18-08015-t002:** Occurrence of AFB_1_ and total AFs in edible vegetable seed samples from imported and local origin.

Sample Category	Local Samples							
	Total Sample	Positive Samples	AFB_1_	Total AFs	Total Samples	Positive Samples	AFB_1_	Total AFs
	N	N (%)	Mean (µg/kg) ± S.D.	Mean (µg/kg) ± S.D.	N	N (%)	Mean (µg/kg) ± S.D.	Mean (µg/kg) ± S.D.
Sunflower	25	18 (72.0)	13.29 ± 3.5	20.428 ± 5.2	32	20 (62.5)	14.23 ± 3.6	20.27 ± 5.4
Soybean	15	7 (46.6)	10.36 ± 4.1	18.42 ± 4.6	12	9 (75.0)	21.01 ± 4.7	36.37 ± 6.1
Canola	28	14 (50.0)	9.30 ± 2.4	16.23 ± 3.7	34	18 (52.9)	13.29 ± 5.6	20.05 ± 5.9
Olive	22	8 (36.3)	5.74 ± 2.5	9.48 ± 1.6	20	10 (50.0)	11.24 ± 2.5	16.50 ± 5.4
Corn	38	25 (65.7)	10.16 ± 4.3	16.08 ± 2.9	45	28 (62.2)	12.80 ± 5.3	19.92 ± 3.5
Mustard	34	20 (58.8)	6.41 ± 2.7	12.95 ± 3.5	46	23 (50.0)	8.74 ± 4.5	13.88 ± 4.6
Total	162	92 (56.7)			189	108 (57.0)		

The data in parenthesis represent the percentage of total analysed samples; LOD = limit of detection.

**Table 3 ijerph-18-08015-t003:** Occurrence of AFB_1_ and total AFs in vegetable oil samples from imported and local origin.

Sample Category	Local Samples							
	Total Sample	Positive Samples	Mean AFB_1_	Mean AFs	Total Samples	Positive Samples	Mean of AFB_1_	Mean of AFs
	N	N (%)	(µg/kg) ± S.D.	(µg/kg) ± S.D.	N	N (%)	µg/kg ± S.D.	µg/kg ± S.D.
Sunflower	20	12 (60.0)	5.93 ± 2.3	11.16 ± 2.9	25	15 (60.0)	8.7 ± 3.2	15.1 ± 4.3
Soybean	18	9 (50.0)	6.94 ± 1.9	12.71 ± 4.3	15	10 (66.0)	14.29 ± 2.5	25.61 ± 7.5
Canola	42	18 (42.8)	4.87 ± 1.8	9.25 ± 3.4	50	22 (44.0)	7.41 ± 3.4	11.80 ± 3.5
Olive	20	8 (40.0)	4.38 ± 2.4	7.47 ± 2.4	18	8 (44.0)	8.51 ± 3.5	12.78 ± 5.3
Corn	45	10 (22.0)	2.43 ± 1.9	3.98 ± 1.5	55	20 (36.3)	6.34 ± 2.8	8.83 ± 2.8
Mustard	35	21 (60.0)	5.69 ± 2.4	11.56 ± 1.9	50	28 (56.0)	7.71 ± 1.8	13.43 ± 4.6
Total	180	78 (43.3)			213	103 (48.3)		

The data in parenthesis represents the percentage of total analysed samples; LOD = limit of detection.

**Table 4 ijerph-18-08015-t004:** Estimation of dietary intake (µg/kg/day) for total AFs in sunflower oil samples.

Category	Type	Males			Females		
		Age Groups			Age Groups		
		16–22	23–32	≥33	16–22	23–32	≥33
Imported	Consumption mg/day	2.7	2.2	2.1	2.5	2.1	1.9
	AFs mean level (µg/kg)	11.16	11.16	11.16	11.16	11.16	11.16
	Dietary Intake µg/kg/day	0.49	0.32	0.25	0.57	0.43	0.33
Local	Consumption mg/day	2.8	2.4	2.1	2.9	2.2	2.0
	AFs mean level (µg/kg)	15.1	15.1	15.1	15.1	15.1	15.1
	Dietary Intake µg/kg/day	0.69	0.48	0.33	0.90	0.58	0.45

Exposure of dietary intake = mean contamination level × per capita consumption/average weight.

## Data Availability

The data will be available, when requested.

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
