# Peer review of "Occurrence of Aflatoxins in Edible Vegetable Seeds and Oil Samples Available in Pakistani Retail Markets and Estimation of Dietary Intake in Consumers"

_ijerph, 2021, doi:10.3390/ijerph18158015_

Round 1

Reviewer 1 Report

This manuscript presents results related to AFB1 and Total Aflatoxins concentrations (AFB1+AFB2+AFG1+AFG2) in edible seeds and vegetable oils.

The occurrence of aflatoxins in these food products (n=744) and the estimation of dietary intake by applying a Food Frequency Questionnaire (n = 589) are valuable data and are worthy of publishing.

However, the manuscript contains some confusing paragraphs (English style and spelling) and a lack of information that might depreciate its relevance. 

Below are some points that authors might consider:

1) 2.3. Extraction of aflatoxins from seed samples

a) 125 mL of methane: water- or 125 of methanol: water?

b) Whitmanpaper – or Whatman paper? 

c) The lower 25 mL of aqueous methanol phase– It seems 125 mL of aqueous methanol was added. What happened with the remaining volume?

2) 2.4. Extraction of aflatoxins from oil samples

Derivatization step with TFA was used only for oil samples or also for seeds? It is not clear from the procedure described. 

3) 2.5. HPLC conditions

Excitation and Emission wavelengths, described by AOAC official methods, are usually around 365 nm and 450 nm, respectively, when using TFA precolumn derivatization. Is there any specific reason for using 295 nm (excitation) and 325 nm (emission)? 

4) 3.1. Quality control parameters

It is not mentioned the number of replicates for each parameter: 

Recovery - how many times were extractions performed for each fortified level? There is a Relative Standard Deviation (RSD) related to it. How many injections for each solution to calculate Standard Deviation for migration time? 

Repeatability and Reproducibility – Specify the number of replicates (n = ?). What were the criteria used for these two terms (intra-day and inter-day)? These definitions must be unambiguous in the procedure. 

5) 3.3. Estimation of dietary intake in sunflower oil samples

Calculation of Dietary Intake also for AFB1 separately would enrich the discussion.

Reviewer 2 Report

This study investigates the occurrence of aflatoxins in vegetable seeds and oils in Pakistan. I consider it to be of interest to those in the field.

Specific comments are provided below:

ABSTRACT

  1. The abstract should be restructured. Please include the following sections in the following order: Background, Methods, Results, and Conclusions; please reduce the results accordingly to that presented in the ‘highlight for review’ section.

INTRODUCTION

  1. The introduction is a little scattered and some important concepts should be mentioned to contextualize the reader. I suggest structuring it in the following topic order (citing both the concepts and theoretical and/or practical examples): vegetable oils (what are they and their culinary uses; consumption of edible vegetable oils worldwide and in Pakistan; their potential advantages and long-term health effects); aflatoxins (concept and their health implications; their occurrence in edible vegetable oils (how they occur and the implications of their occurrence for the public health); the studies performed in this field (scientific literature available concerning vegetable oils and the occurrence of aflatoxins) and the gap of studies concerning this subject in Pakistan (which should also be used to provide the justification for your topic research). One paragraph for each topic should be enough to contextualize and you may use the information you already have.
  2. Please check the information included in the aim of your study. In the results section, the concentrations of two aflatoxins (AFB1 and AFG1) are mentioned; nonetheless, only aflatoxin B1 is mentioned on the aim of the study in the introduction section.

MATERIALS AND METHODS

  1. I suggest inserting the following information (“The branded samples were named those imported from other countries, and non-branded were produced lo-cally. These branded sunflower seeds samples were collected from 5 different brands, 7 different brands for soybean, 8 different brands of canola, olive and corn samples and 4 brands for mustard samples main cities and subsequently the oil was extracted from these samples and labeled accordingly (Lahore, Fai-salabad, Gojra and Islamabad) of Punjab Pakistan”) into a hierarchy SmartArt graphic, for example, to ease reader’s understanding.
  2. There is little information regarding the sample of the study. How were the edible seeds and locations selected? What was the calculation or method for determining the sample size? When the collection process took place? Were there any criteria for the selection of samples? How much the collected data can be considered representative, or at least significant, in relation to the studied topic?
  3. Please clarify/justify the selected methods used for the determination of aflatoxins in samples (what were the rules established for the choice of the methods used, if those methods meet the scientific standards and are in accordance with the official methods used for this purpose, etc.).
  4. Were the ethical issues concerning a survey-based research addressed? Please add the information about the ethical approval of the research involving human participants. There is also little information concerning the participants of the study. How were the participants selected? What was the calculation or method for determining the sample size? When the collection process took place? What were the inclusion and exclusion criteria of participants? How much the collected data can be considered representative, or at least significant, in relation to the studied topic? There is also little information about the questionnaire. Why was it chosen? Does it have any kind of validation protocol? How were they applied to participants (self-administered, interview, etc.)? How long it took to be completed? When was it applied? How were the aspects of oil consumption, dietary supplements, and utilization assessed? and other relevant information.
  5. It is unclear what was/were the guideline/s or scientific criteria used to determine the aflatoxins limit of the samples assessed. Please identify the regulations and/or standards used and their respective proposed/permitted limits.
  6. Please include the probability value (p-value) adopted in the statistical tests performed.

RESULTS AND DISCUSSION

  1. Please provide the identification of all the codes and acronyms at least once throughout the manuscript (i.e.: AFB1, AFG1, AFB2, AFG2, EU, OTA and others which may apply).
  2. Please change the sentence “The data in parenthesis represents the percentage to total analysed samples” to “The data in parenthesis represents the percentage of total analysed samples”.
  3. However, due to uneducated and no knowledge of the implications of toxigenic mold growth, the farmers, processors, and food han-dlers make these crops' quality questionable”: Please reformulate this sentence.
  4. The content of the “Results and Discussion” section is almost exclusively addressed to the results found in other studies. You might consider explaining the significance of your findings. What does these results mean? What are the possible explanations for these findings? You might also consider citing the methods adopted in other studies when comparing them with the results of your studies, since different methodologies may lead to diverse or conflicting results. The following may be helpful to restructure the discussion of your manuscript:
  • Statement of principal findings;
  • Strengths and weaknesses of the study;
  • Strengths and weaknesses in relation to other studies, discussing particularly any differences in results;
  • Meaning of the study: possible mechanisms and implications for researchers in the field;
  • Unanswered questions and future research.

Reference: Docherty, M., & Smith, R. (1999). The case for structuring the discussion of scientific papers. BMJ (Clinical research ed.), 318(7193), 1224–1225. https://doi.org/10.1136/bmj.318.7193.1224

Also, some parts of the discussion should be revised to emphasise the potential implications for future intervention. What are the lessons learnt? How will the findings apply to further the field?

The same observations apply to the results of the estimation of dietary intake assessed with consumers. What are the lessons learnt? How will the findings apply to further the field? What are potential implications of this study and its results (estimation of dietary intake of AFs) from the public health perspective? What kind of interventions should be provided (for policy makers/regulatory agencies and consumers)? What points should be addressed in future research?

CONCLUSIONS

  1. What are strategical implications of this study for businesses and policy makers? What kind of interventional strategies should be provided to farmers, processors, and food handlers workers in this field? What points should be addressed in future research?
  2. You might consider creating a limitation section, adding the advantages and limitations in aflatoxins analysis methods and the potential solutions for overcoming these obstacles.

Author Response

The response has attached

Reviewer 3 Report

The work presents the study of a consistent number of data collected from the analysis of different vegetable seeds and oils to determine the amount of aflatoxins. In addition, an estimation of dietary intake is proposed.

First of all, the comprehension of the work is difficult, due to a wrong use of the English.  Therefore the grammar, terms and syntax of each section need a deep revision and correction.

The Abstract is a mass of data which creates confusion, without giving a clear idea of the aims and results. It must be rephrased, including the state of the art, the aim, the most important results obtained and the conclusions.

In my opinion the use of the term "branded" and "non-branded"  is questionable since can be confused one another. I suggest the use of imported and local, respectively.

Highlight for review is a repetition and has no sense, should be removed.

The Introduction must be improved:

  1. I suggest to put as first the sentence starting "The contamination of food...⌊17,18,1⌋".
  2. Add a description of the mycotoxins family, which are further present in Table1 and add a description of the other AFs which are mentioned in the Results and discussion.
  3. Which are the current regulatory limits for AFs in Pakistan? Since the European Union is mentioned, please insert the recommended limits. 
  4. Are there any approved limits for humans intake?
  5. Please check and ally the brackets for References.
  6. Please, rephrase or explain what the authors wants to say with the sentence "The findings will help understand the toxicity of AFs....".
  7. Enlarge the Introduction by adding some studies on this topic.

In Materials and Methods should be explained better:

  1. Describe how the oils were extracted from the samples and by which instruments. These mentioned extraction of oils creates a misunderstanding, are they used for the analysis?
  2. Clarify the sentence “The sample size was confirmed at their packaging sizes and varies between 1 kg, each.”
  3. The specification of the laboratory can be removed.
  4. Check and complete the chemicals and reagents including: the list of the AFs standards and concentrations , the purity grade of solvent. Moreover, it seems that HCl is missing in the list, please check.
  5. Check the brackets for references, acronyms, the use of present or past form.
  6. Rephrase the extraction of AFs from seed samples and clarify how is possible to collect the lower aqueous phase after filtration. Check the solvent: methanol and not methane. Do you have any idea how many times the extraction process with chloroform is repeated?
  7. For the extraction of AFs from oil samples, which is the AOAC method?  
  8. Add v/v for methanol-water, correct HCL as HCl and convert the rpm in g.
  9. Since the derivatization reaction is common for seeds and oils, separate its description with a new paragraph.
  10. Enlarge the HPLC conditions describing which AFs were analyzed, which are the chromatographic conditions (isocratic or gradient). Check the mobile phase, is it correct the amount of acetic acid? Move close to the fluorescence detector the wavelengths.
  11. For Dietary intake estimations the supplementary materials are not present, therefore the evaluation of the pertinence of the survey on dietary is impossible. Please add.
  12. Replace can be calculated with is calculated and in the equation explain the average weight (of who?)
  13. Describe in this paragraph the age and sex of the 645 participants.
  14. Add Validation to the title of Statistical analysis. What is average referred to? Insert the levels used for the calibration curve.

For Results and Discussion must be explained better:

  1. Please explain the acronyms for AFG1, AFB2 and AFG2 and according to what did you chose those levels (1,6,10 µg/L and 1,4,8µg/L). Please verify the concentrations expressed in Table 1, which is µg/kg differently from here.
  2. It is not clear how the authors added the standards to a mix sample of edible oil, please rephrase it.
  3. In my opinion, the Discussion should start with the sentence “ The mean recovery values…”. All written before should be moved to Statistical analysis, as well as the sentence about the calibration curves. Check the LOQ values for AFB2 and AFG2, is 0,21 or 0,021 μg/kg?
  4. It is not clear what the authors want to say in relation to the previous study.
  5. Change the title of Table 1, replacing recovery analysis with validation parameters.
  6. Repeatability and reproducibility are not mentioned in the text, please introduce a comment. Add RSD% to the table.
  7. Remove the title of figures from the text and anyway, rephrase it.
  8. The sentence “The research was conducted to investigate…central cities of Punjab, Pakistan” is redundant, please remove it.
  9. Check (also in the Materials) and align the numbers to one decimal.
  10. Clarify if the oil samples are 393 or 392.
  11. There is no accordance between what written in the text and the relative reference to the Figure, please rephrase better the explanation of the data.
  12. It is not clear, in the end, which one is more contaminated? Is it the branded or non-branded? Do the authors have an hypothesis why the levels are so high?
  13. Please insert the results for t-test and check p, that should be 0,0095.
  14. Regarding Tables 2-3, Range column can be removed. Moreover, the maximum level is quite different from the mean. Please add to the supplementary data, all the values obtained for each sample analyzed.
  15. The levels of EU limits are reported in different ways, please align to μg/kg.
  16. Include, for each reference mentioned the value of LOD when indicated.
  17. What is OTA?
  18. Rephrase the last part of the discussion, it is not clear.
  19. For the Estimation of dietary intake, put the second sentence (regarding the reason why sunflower oil was chosen) as first.
  20. Are there any regulations/normative about the intake?
  21. Is ≤33 correct? Would you mean ≥33?
  22. The consumption ml/day mentioned in Table 4, is not present in the text.
  23. What is exposure referred to?

Regarding Conclusions, please rephrase it, including the reason why this study was conducted and its utility, future perspectives. Do you want the Pakistan authority to evaluate this study for regulatory purposes?

In conclusion, the entire work needs and intensive English revision with rephrase, therefore consider the above points as major revision.

Author Response

Attached file

Round 2

Reviewer 2 Report

Thank you. This is an improved version of your manuscript.

However, the suggestions were only superficially addressed in the conclusion; therefore, this section still needs improvement. Please address the following:

  1. What are strategical implications of this study for businesses, policy makers and - especially - consumers? What kind of interventional strategies should be provided to farmers, processors, food handlers workers, and even consumers, in this field? What points should be addressed in future research? You might benefit from consulting similar studies and their conclusions to help you build yours.

  1. Please remove the following sentence (“However, liquid-liquid extraction instead of immunoaffinity columns and unavailability of LC-MS analysis are the weak point of the present study”.), since it is not the only limitation of your work. Please add a limitation section, separately from the conclusion or any other section, and mention the advantages and limitations of your work as a whole and the potential solutions for overcoming these obstacles.

Author Response

The response has attached

Reviewer 3 Report

The revision of the entire work has improved the whole manuscript, in terms of easily of reading, interest for scientists and for the involved authority. Thanks for the determined approach.

Anyway, some minor aspects can be changed.

The abstract is fine, but I advice to check:

  • the p value of t-test.
  • the decimal part of % used in the text. What I mean is, if you decide to write, for example: The results have shown that 92 ( 56.7%) <with 1 decimal> samples of imported and 108 (57%) <no decimal used, please add 1 decimal as well as above, I guess it's ,0>. Sometimes the % is expressed, in the text, with one decimal, sometimes with no decimal. Therefore, please uniformy this aspect for all the manuscript.

Regarding Highlight for review, I only suggest to remove but if you want to leave it there you can do it, but please don't reply, in the cover letter, that it has been removed, if it's not.

The introduction is fine, just some check:

  • remove the repitition of vitamins inside the brackets (vitamins A, D and E)
  • add the acronymous FAO
  • The sentence : In tropical countries...is repeated twice.

Materials and methods is fine but please check Fig. 1, for olive and corn how many different brands were collected? 8 as in the text or 4 as in Fig.1?

Also for Results and discussion, please check

  • Figure 2, under the frigure the text mention flour AFs instead of four AFs.
  • Figure 3 a) and b), branded/ non branded is still present. Moreover:
  • for Fig. 3 a) move up the legend text as it cover the value of sunflowers column.
  • for Fig. 3 b) check the value added because are on the wrong column (<20 µg/kg)

Regarding the conclusion, I suggest to clarify better what the authors want to state witht the last sentence on the weak point of the present study.

In the end, check the space between words as sometimes are attached (oilseeds or inedible), plural words (tons not tones) uniformy units (mL or ml).

After the above minor revision the text will be a good manuscript and therefore can be accepted.

Author Response

The response has attached
